# Comparative Flavor Profile of Roasted Germinated Wheat (*Triticum aestivum* L.) Beverages Served Hot and Cold Using Electronic Sensors Combined with Chemometric Statistical Analysis

**DOI:** 10.3390/foods11193099

**Published:** 2022-10-05

**Authors:** Thinzar Aung, Bo Ram Kim, Mi Jeong Kim

**Affiliations:** 1Department of Food and Nutrition, Changwon National University, Changwon 51140, Korea; 2Interdisciplinary Program in Senior Human Ecology, Changwon National University, Changwon 51140, Korea

**Keywords:** electronic nose, electronic tongue, germination, bioactive components, cereal beverage, volatile compounds

## Abstract

In order to fulfill the objective of the comparative flavor profiling of hot and cold serving, different concentrations of hot (hot infusion) and cold (boiled and cold serving) roasted-steamed-germinated wheat beverages were prepared in order to explore the comparative profile of the amino acids, volatiles, taste, total flavonoid content (TFC), total phenolic content (TPC), and antioxidant capacities, including 2,2-diphenyl-1-picrylhydrazyl radical scavenging activity (DPPH) and the Trolox equivalent antioxidant activity (TEAC). The instrumental analysis was performed using electronic sensors (an electric nose and an electric tongue), high-performance liquid chromatography, and spectrophotometry, and the statistical assessment was conducted using univariate (correlation pattern and heatmap) and multivariate (clustering and principal component analysis) analyses. The cold beverages at the highest concentration showed the highest values of TFC, TPC, DPPH, and TEAC, showing 32.31 ± 1.13 μg CE/100 mL, 202.37 ± 20.94 μg GAE/100 mL, 68.43 ± 3.41 μM TE/100 mL, and 126.66 ± 4.00 mM TE/100 mL, respectively. The correlation analysis revealed a remarkable correlation between the taste and the flavor compounds. The clustering analysis and the PCA clearly divided the key metabolites, which were attributed to the different tastes in the hot and cold beverages. This study clearly demonstrates the impact of different temperatures on the aroma metabolites, the taste, and the characteristics of wheat beverages.

## 1. Introduction

The awareness of a healthy lifestyle and the consumption of “functional drinks” have increased in order to improve wellness and prevent the risk of diseases [1,2]. This concept of “functional drinks” claims that they possess health benefits, as they serve as an ideal delivery vehicle for nutrients and other functional substances, such as flavonoids, specific amino acids, antioxidants, minerals, vitamins, fibers, and phytonutrients, in a convenient way in order to fulfill the customers’ demands [3]. Cereal-based beverages that are prepared from cereals such as barley, oats, rice, wheat, and pseudocereals, such as buckwheat and quinoa, are considered to be functional beverages in a broader sense [1]. Wheat (*Triticum aestivum* L.), which is a major cereal grain, is associated with potential health benefits, including antidiabetic, anti-inflammatory, and anticancer properties, owing to the presence of basic nutrients, such as proteins, carbohydrates, vitamins, and bioactive components, such as phenolic acids, flavonoids, and anthocyanins [4,5]. The potential use of wheat as a novel cereal-based beverage has already been introduced in our previous research, along with the health-related properties of wheat during different beverage preparation steps [6].

In designing a drink that is formulated with cereals, it is important to consider the feasible processing steps, as most of the parts of cereal grains, such as the proteins and the cell wall polysaccharides, are insoluble, and the yield for the drink is insufficient [1]. In addition, the raw material selection, the processing techniques, the sensory profile, and the functional characteristics are key factors in achieving a safe, high-quality, and marketable product [7]. In grain processing technology, germination and roasting are advantageous processes for providing the positive aroma and flavor components of Maillard reactions, reducing the undesirable flavor and aromatic components, hydrolyzing the starch, the proteins, and the cell wall polysaccharides by activated enzymes, and enhancing the functional components, as well as the phenolic compounds [1,8]. In a previous study, wheat was prepared by germinating, steaming, and roasting for beverage preparation, and the quality characteristics and sensory properties were explored in each preparation step [6]. In addition, the germinated wheat clearly exhibited a higher level of bioactive components, such as *γ*-aminobutyric acid, phenolic compounds, and flavonoid compounds, than the wheat, due to the activation of the related enzymes during germination [6].

In addition to utilizing the different preparation steps in order to enhance the functional components and antioxidant capacities of wheat beverages, it is necessary to consider the effect of the serving types, such as hot infusion or cold serving, on the food components, the taste attributes, the nutritive value, and the functional properties of the beverages. The serving of hot and cold beverages might interact or alter these components, owing to the use of different temperatures and the loads of the raw materials that are used. Moreover, further experiments are required in order to understand the possibility of the full use of these functional components and the intensity of the aroma and flavor profiles in roasted-steamed-germinated-wheat beverages at the serving stage. The current study fills this gap by exploring the profile of the amino acids, the volatiles, and the taste of previously optimized roasted-steamed-germinated-wheat beverages at hot and cold serving types, accompanied by their bioactive components and antioxidant activities, using electronic sensors, including an electric nose (E. nose) and an electric tongue (E. tongue), high-performance liquid chromatography, and spectrophotometry. There are no studies comparing the various components of a product, such as a beverage, which may be sensitive to temperature when it is consumed. Regarding the electrochemical detection method, the sensors, e.g., the E. nose and the E. tongue, which attempted to imitate human smell and taste, were applied in order to achieve the effective sensory attributes [9]. Thus, the aim of this study was to compare the flavor profiles of hot and cold wheat beverages, with respect to their nutraceutical potential, using chemometric statistical analysis.

## 2. Materials and Methods

### 2.1. Raw Materials and Chemicals

Wheat seeds (*Triticum aestivum* L. var. *Anzunbaengi*) were purchased from Jinju, Republic of Korea. The following chemicals: gallic acid (PubChem CID: 370), catechin (PubChem CID: 9064), Trolox (PubChem CID: 40634), 2,2-diphenyl-1-picrylhydrazyl (DPPH) (PubChem CID: 129737978), and 2,2′-azino-bis (3-Ethylbenzothiazoline-6-Sulfonic acid) (ABTS) (PubChem CID: 16240279) were purchased from Sigma-Aldrich (St. Louis, MO, USA).

### 2.2. Sample Preparation

The germination of the raw wheat was carried out at 17.6 °C for 46.18 h after soaking in distilled water (1:5 *w*/*v*) at 25 ± 2 °C for 12 h [10]. The germinated wheat was dried in an oven at 45 °C for around 12 h, was steamed at 220 °C for 10 min, and was roasted in a roaster (MK-300, JC, Seoul, Korea) at 180 °C for 44.56 min continuously [6]. The roasted-steamed-germinated wheat (RSGW) was used for the preparation of hot and cold beverages [6]. Regarding the hot beverages (HB), different amounts (0.8, 2, and 4 g) of ground roasted-steamed-germinated wheat were placed in a tea bag and infused in 100 mL of boiling water for 25 min, and each beverage was labelled as HB_1, HB_2, or HB_3, respectively. For the preparation of the cold beverages (CB), 20, 50, and 75 g of RSGW was mixed with 1 L of water, was boiled for 30 min, was cooled in a refrigerator at 4 °C, and then labelled as CB_1, CB_2, or CB_3, respectively.

### 2.3. Analysis

#### 2.3.1. Nutraceutical Characteristics of Roasted Germinated Wheat Beverages

Prior to the analysis of the total phenolic and flavonoid contents, 80% ethanol was used to prepare the extract using a shaking incubator at 100 rpm for 3 h at 65 °C. Extracts were collected by filtering with a 0.45 µM filter after centrifuging at 4000 rpm for 10 min, and were then stored at −20 °C. The total flavonoid content (TFC) was determined using a spectrophotometric method [11]. A mixture of the extract (100 μL), distilled water (1.25 mL), and 5% sodium nitrite (75 μL) was kept for 6 min and then 10% aluminum chloride (150 μL) was added. After 5 min, 1 M of sodium hydroxide (0.5 mL) was added to the mixture, and the TFC was determined at 510 nm using a spectrophotometer (EMC-11D-V, EMCLAB Instruments, Duisburg, Germany). Catechin was used as the standard and the values were shown as mg CE/g sample. The total phenolic content (TPC) was analyzed using the method of [12]. The sample extract (20 µL) was reacted with the reagents, 20% sodium carbonate, and 10% Folin–Ciocalteu reagent, in the dark for 2 h at 25 ± 2 °C. The TPC was measured at 765 nm using a spectrophotometer (EMC-11D-V, EMCLAB Instruments, Duisburg, Germany), and the results were displayed in mg GE/g of sample using a gallic acid equivalent standard.

For DPPH radical scavenging activity, the method described by [13] was used. Briefly, the absorbance of a mixture (50 μL extract, 1950 μL 0.1 mM DPPH), which was incubated in the dark for 30 min at 25 ± 2 °C, was detected using a spectrophotometer (EMC-11D-V, EMCLAB Instruments, Duisburg, Germany). Trolox was used as a standard, and the values were expressed as µM TE/g sample. For the Trolox equivalent antioxidant activity (TEAC), the spectrophotometric method reported by [14] was used. The ABTS solution (7.4 mM ABTS and 2.6 mM potassium persulfate), which was kept in the dark for 16 h, was diluted with 100% methanol until an absorbance value of 0.7 at 734 nm was reached. Then, 2980 µL of the previously diluted ABTS solution and 20 µL of the sample extract were mixed at room temperature for 7 min. The absorbance was measured at 734 nm using a spectrophotometer (EMC-11D-V, EMCLAB Instruments, Duisburg, Germany), and the results were expressed as mM TE/g sample.

#### 2.3.2. Amino Acid Profile of Roasted Germinated Wheat Beverages

The free amino acids were determined by HPLC (e2695, Waters Corp, Milford, MA, USA) with a PDA detector (e2998, Waters Corp, Milford, MA, USA) using the Waters HPLC AccQ-Tag method [15]. Briefly, the sample (1 g) was hydrolyzed with 0.1 N HCl (5 mL) for 15 min and then centrifuged at 8000 rpm for 15 min. After centrifugation, the supernatant was filtered through a 0.45 μM filter and used for analysis. The column (3.9 × 150 mm) was set at 37 °C and the sample temperature was set at 20 °C. The solvents used for the mobile phase were 10% acetate phosphate buffer (Eluent A, WAT052890, Waters Corp, Milford, MA, USA), HPLC-grade acetonitrile, and HPLC-grade water. The standard amino acids (amino acid standard H, Waters Corp, Milford, MA, USA) used in this study were as follows: L-alanine, L-arginine, L-aspartic acid, L-cystine, L-glutamic acid, glycine, L-histidine, L-isoleucine, L-leucine, L-lysine, L-methionine, L-phenylalanine, L-proline, L -serine, L-threonine, L-tyrosine, and L-valine.

#### 2.3.3. Volatiles Profile of Roasted Germinated Wheat Beverages

An electronic nose (HERACLES Neo, Alpha MOS, Toulouse, France) was used to identify the volatile compounds of hot- and cold-roasted germinated wheat beverages. The wheat beverage samples (3 g) were placed in a 20 mL vial and heated at 50 °C for 10 min. Approximately 5 mL of each sample was collected and injected at 200 °C into the GC system. Each vial was stayed in a trap for 50 s at 40 °C and then the volatile compounds that were carried by hydrogen gas were passed to the column. The temperature was increased from 50 °C to 250 °C at a rate of 1 °C/s (up to 80 °C) and 3 °C/s (up to 250 °C). Kovat’s indices were used to identify the peaks of the samples, and the volatiles were confirmed based on the AroChemBase library.

#### 2.3.4. Taste Profile of Roasted Germinated Wheat Beverages

An Astree electronic tongue system (Alpha M.O.S, Toulouse, France) was used to test the taste traits of the roasted germinated wheat during hot and cold preparation. A sample stock solution (100 mL) was used for analysis. Taste component sensors (SRS-sourness, STS-saltiness, UMS-umami, SWS-sweetness, BRS-bitterness), indicator sensors (GPS-metallic, SPS-spiciness), and a reference electrode (Ag/AgCl) were used in the system. After placing the sample in the E-tongue sampler, the sensor was immersed in the sample for 2 min, and the intensity of each taste sensor was measured by dipping. To prevent errors and contamination between the samples, each taste component sensor was rinsed with distilled water before a new measurement cycle. The results were confirmed as taste patterns for the taste components.

### 2.4. Statistical Assessment

For the analysis of the physical characteristics, including solid content, yield, and color properties, and the nutraceutical characteristics, including TPC, TFC, DPPH, and TEAC; amino acids; volatile compounds; and taste profile by E-nose, triplicate samples were determined, expressing as mean ± standard deviation. For the statistical assessment of each parameter, analysis of variance (ANOVA) with Duncan’s multiple comparison test as a post hoc test was performed by comparing the significant differences in the mean values between the samples (α = 0.05). The clustering analysis and correlation analysis, including pattern assessment and a Pearson correlation heatmap between the aroma metabolites and characteristics of hot and cold beverages, were performed using MetaboAnalyst (https://www.metaboanalyst.ca, accessed on 19 August 2022). The principal component analysis (PCA) of the nutraceutical properties, amino acid compositions, volatile compounds, and taste profiles of the hot and cold beverages was performed using XLSTAT (ver. 2021.2, Addinsoft, Paris, France).

## 3. Results and Discussion

### 3.1. Nutraceutical Characteristics of Roasted Germinated Wheat Beverages

The nutraceutical properties of the RSGW beverages showed a significant difference depending on the type of preparation and the amount of samples that were used for each method. The total phenolic and flavonoid content, the DPPH radical scavenging activity, and the TEAC values of the hot and cold beverage samples are shown in Figure 1 and Appendix A. Generally, the hot beverages showed lower TFC and TPC contents, as well as antioxidant activities, compared to the cold samples, due to the amount of solvent that was used for brewing. In the cold beverages, it can be clearly seen that the greater the concentration of RSGW in the preparation process, the higher the level of bioactive components and antioxidant capacities. The content of TFC and TPC in the beverage samples was related to the formation of thermally induced products by the roasting of the germinated wheat samples, which, in terms of the Maillard reaction, generated bioactive compounds with higher antioxidant potential [6]. This statement on the enhancement of bioactive components, as well as the antioxidant capacities by the germination and roasting of wheat, has already been proved in our previous study [6]. Among the cold wheat beverages, the highest level of TFC and TPC was observed in CB_3, amounting to 32.31 ± 1.13 μg CE/100 mL and 202.37 ± 20.94 μg GAE/100 mL, followed by CB_2 (2.34 ± 1.39 μg CE/100 mL and 160.42 ± 19.10 μg GAE/100 mL), and CB_1 (10.86 ± 0.47 μg CE/100 mL and 58.25 ± 4.43 μg GAE/100 mL). The highest value of DPPH and TEAC were observed in CB_3, showing 68.43 ± 3.41 μM TE/100 mL and 126.66 ± 4.00 mM TE/100 mL, respectively. A slight variation in the increasing trend for both the bioactive components and the antioxidant activities was found in the hot samples, in contrast to the cold samples. The highest level of these parameters was detected in the HB_2 sample, which contained 2 g/100 mL of RSGW, instead of showing the highest values at the highest concentration of RSGW. This may be related to the kinetics of infusion, due to the use of tea bags in the preparation of hot beverages. In this study, tea bags of the same size were used for the infusion of the RSGW powder at different concentrations. Thus, the surface area of the bag with the higher amount of RSGW might be small, and the space may have become compact together by swelling after infusing it in hot water; moreover, this showed the highest solid content and yield in the HB_2 sample (Appendix A). It is more difficult to transport the bioactive components through the Nernst diffusion layer to the RSGW in the center of the tea bag, resulting in a slow infusion process [16].

### 3.2. Amino Acid Profiles of Roasted Germinated Wheat Beverages

The amino acid profiles of the roasted germinated wheat beverages under cold- and hot-brewing conditions are listed in Table 1. The following: alanine (Ala), arginine (Arg), aspartic acid (Asp), glutamic acid (Glu), glycine (Gly), histidine (His), isoleucine (Ile), leucine (Leu), lysine (Lys), methionine (Met), phenylalanine (Phe), proline (Pro), serine (Ser), threonine (Thr), tyrosine (Tyr), and valine (Val) were detected. The major contributing factors of amino acids in RSGW beverages are the raw material processing steps, including the germination, the steaming, and the roasting, which promote the amino acid content in the samples by activating the proteolytic enzymes to liberate the peptides and free the amino acids [17]. When comparing hot and cold brewing, a relatively high concentration of amino acids was exhibited in the cold beverages, as well as Met, which was only detected in the cold beverages with higher brewing concentrations. This might be the reason for the higher concentration of RSGW that is used in the cold brewing process than in that of hot beverages. Under hot-brewing conditions, a decrease in amino acids was observed in the tea bags with a greater amount of RSGW. A similar study has reported that a lower amount of loading in tea bags resulted in a higher permeability of the tea bags, showing higher infusion kinetics by increasing the volumetric mass transfer coefficient [18]. In this case, it is important to consider the infusion kinetics during the leaching process (solid–liquid extraction), which largely depends on the infusion temperature, the particle size, and the loading in the tea bags [18]. Yadav et al. (2017) stated that the infusion process includes the swelling of tea granules and tea bags, which increases the compactness of the tea bags and reduces the infusion kinetics, resulting in lower gallic acid infusion [19]. Thus, during infusion, the compressed tea bag can cause the hinderance of the transfer of amino acids [18].

From a sensory perspective, amino acids contribute to the taste of food by interacting with other food constituents, such as nucleotides and inorganic salts; although, amino acids themselves do not have a strong taste profile [20]. In hot beverages, branched-chain amino acids (BCAAs), such as Ile, Leu, and Val, which exhibited the bitter taste profile [21], showed a significantly lower level (*p* < 0.001) in the higher amount of infusion concentration, amounting to 1.90–0.24 mg/100 g for Ile, 0.65–0.43 mg/100 g for Leu, and 1.01–0.69 mg/100 g for Val, respectively. Although a slight increase in BCAAs was observed, the values showed no significant differences in the concentrations of wheat in the cold beverages. Other bitter amino acids, such as Phe and Tyr, also displayed significant decreases in HB_2 and HB_3. The amounts of Pro and Ser were not significantly different for both the hot and the cold beverages, and they lost their bitterness in the solution form [21]. Lys and Thr, which provide a sweet taste, were significantly increased in higher amounts of wheat in the cold beverage preparation, showing the highest values (Lys, 3.52, and Thr, 2.82 mg/100 g) in CB_3. The amino acids with an umami taste, such as Glu and Asp, decreased significantly in the cold beverages with higher concentrations of wheat, but showed no significant difference in the hot beverages.

### 3.3. Volatiles Profile of Roasted Germinated Wheat Beverages

The aroma metabolites of RSGW beverages in the different preparation types were quantitatively distinguished with an electronic nose, and 24 volatiles, including acids and esters (5), alcohols (5), aldehydes (2), and hydrocarbons (9), were detected in the hot and cold wheat beverages (Table 2 and Figure 2). Generally, the amount of wheat for infusion in the hot beverages showed no significant differences between the volatile compounds, except for methyl formate, butylate, 3-pentanol, 2,3-dimethyly-5-ethylpyrazine, and 2-methylpentane, which displayed significant differences (*p* < 0.05). HB_2 exhibited the highest abundance of volatile compounds, followed by HB_1, and HB_3. For the cold beverages, the level of wheat that was used in brewing significantly affects the abundance of volatiles. The acids and esters that were found in this study were methyl formate, propanoate, pentanoate, benzyl acetate, and butylate. Methyl formate, which is associated with fruity notes, was the main type of ester that was found in the hot beverages, showing the highest level (3.77 ± 0.20) in HB_3. Among the cold beverages, methyl pentanoate, which has a nutty and sweet odor, was found to be the highest in the CB_3 sample, with the highest peak (4.56 ± 0.95). These esters are related to the esterification of acids, especially formic and acetic acids, which are produced by amino acid degradation during the roasting of the germinated wheat [22]. A relatively low abundance of alcohols, with no significant differences between the amounts of infused tea bags, was observed in the hot beverages, with a relative peak area. In contrast, the cold beverages showed significantly different alcohol contents in the samples that were prepared. Among the five alcohols that were detected (3-pentanol, 3-methyl-1-butanol, 2-methyl-1-butanol, 1-hexen-3-ol, and 2-octanol), 2-methyl-1-butanol, which has an alcoholic and sweet odor, was the most abundant alcohol, showing the highest peak area in CB_3. Aldehydes were found at lower abundances in both the hot and the cold beverages. In both the hot and the cold beverages, 2-methylbutanal and 3-fenylpropenal were detected at low levels, which featured burnt and sweet smells. A similar observation of 2-methylbutanl in tea infusions was reported by [23] and indicated the contribution of a musty, malty, and coca-like aroma. Three heterocyclic compounds, 2,3-dimethyl-5-ethylpyrazine (burnt), myristicin (balsamic, mild), and 4-undecanolide (fruity), were detected at low relative concentrations in both of the serving types. Both 2,3-dimethyl-5-ethylpyrazine and 4-undecanolide were found to be volatile constituents in tea infusions because of their characteristic tea aroma [24].

Hydrocarbons were identified as the dominant volatiles in the hot and the cold RSGW beverages in this study because of the high abundance of these compounds and their considerably higher relative peak area. Among the nine detected hydrocarbons, 2-methylpentane was established as the dominant odorant, showing the highest peak area (40.38 ± 3.98 for HB_2). This generation mechanism may be related to the higher thermal diffusivity of 2-methylpentane in a mixture of alkanes at higher temperatures [25]. Furthermore, the toxic and the carcinogenic volatiles, such as furan, and sulfur-containing compounds with a sulfur odor, were not detected after hot infusion and the boiling of RSGW for the hot and the cold beverage preparation, although relatively small amounts of these compounds were detected in the RSGW powder in our previous study.

### 3.4. Taste Profile of Roasted Germinated Wheat Beverages

The taste traits of the hot and the cold RSGW beverage attributes were identified by the E. tongue system, and the response values of five sensors, including SRS_sourness, STS_saltiness, UMS_umami, SWS_sweetness, and BRS_bitterness, with respect to the samples, are illustrated by spider charts in Figure 3. The response values of taste by the E. tongue revealed significant (*p* < 0.001) differences among the different loading levels of wheat in the hot and the cold beverage preparations (Appendix A). Regarding the hot beverages, similar taste patterns were observed in the samples, except for differences in the taste intensities. As shown in Figure 3, HB_2 displayed the higher level of taste intensity for saltiness (8.00 ± 0.08), sweetness (6.90 ± 0.86), and bitterness (7.83 ± 0.05), while HB_3 showed the highest taste intensity for sourness (6.10 ± 0.08) and umami (4.53 ± 0.05). These taste intensities can be related to the infusibility of the amino acids and the volatiles through the tea bags during infusion, due to the high levels of these components in the HB_2 sample. In the cold beverages, different taste patterns were observed among the tested samples. CB_2 and CB_3 demonstrated similar levels of taste profiles, with higher intensities of sourness and umami. In contrast, CB_1 exhibited the highest bitterness (8.43 ± 0.17), sweetness (9.10 ± 0.28), and saltiness (3.37 ± 0.05), with the lowest intensity of sourness (3.07 ± 0.12) and umami (6.47 ± 0.05). This might be related to the significant difference in the composition of the amino acids and the volatiles between CB_1, CB_2, and CB_3 (*p* < 0.05). According to a report by [9], the surface of detection sensors in an E. tongue are composed of artificial lipid membranes for electrostatic or hydrophobic interactions of flavor compounds, which in turn respond to the different tastes, with respect to the components.

### 3.5. Correlation Analysis between Aroma Metabolites and Characteristics of Hot and Cold Beverages

In order to visualize the contributions of the key compounds and their correlation with the taste profile, a pattern correlation analysis was conducted using Pearson’s correlation through the MetaboAnalyst web version (Figure 4). In Figure 4A–E, the top 25 features correlated with the SRS_sourness, the STS_saltiness, the UMS_umami, the SWS_sweetness, and the BRS_bitterness are illustrated, with respect to their correlation coefficients (r), from −1.0 to 1.0, respectively; the pink color indicates a positive correlation and blue color indicates a negative correlation. Regarding the SRS_sourness (Figure 4B), the BRS_bitterness, the STS_saltiness, and the SWS_sweetness were negatively correlated, exhibiting a higher negative correlation coefficient (r > −0.7, *p* < 0.001), while the UMS_umami taste was positively correlated. The bioactive components, such as TPC, TFC, and antioxidant capacities, showed a positive correlation with the sourness level in the hot and the cold beverages. In addition, the sourness was positively correlated with the components of the acids and the ester volatiles, such as benzyl acetate, methyl pentanoate, hydrocarbon volatiles, such as 2-methylbutane, vinyl benzene, 2-methyl-1heptane, and amino acids, such as Lys, Arg, Met, and Gly, showing higher correlation coefficients (r > 0.5, *p* < 0.001). In the case of STS_saltiness (Figure 4C), most of the features were negatively correlated (r > −0.7, *p* < 0.001), and were positively correlated with bitterness and sweetness (r > 0.6, *p* < 0.001). In response to the UMS_umami, only a positive correlation pattern with the strongest correlation coefficients (r > 0.8, *p* < 0.001) was observed in the tested wheat beverages (Figure 4C). Most of the amino acids (Gly, Ala, Arg, Ser, Thr, Lys, Leu, Val, Phe, His, Pro, Tyr, Glu, and Asp) were responsible for the perception of the umami taste of the germinated wheat beverages. In addition, the presence of the flavonoids and the volatiles, such as 2-methylheptane, 2-methyl-1-butanol, methyl pentanoate, benzyl acetate, 2-octanol, and vinylbenzene, had a positive influence on the umami taste. Wang et al. (2020) stated that umami taste is an indicator of amino acids that can regulate a sweet taste, intensify the saltiness, and suppress the sourness and bitterness [26]. Most of the volatile compounds were negatively correlated with sweetness perception (Figure 4D). Two of the bitter amino acids, Ile and Lys, also had an inverse relationship with sweetness (Figure 4E). A sensitivity to bitterness was associated with saltiness and sweetness (r > 0.7; *p* < 0.001). However, some of the amino acids, such as Ala, Phe, Gly, Arg, and Lys, are negatively associated with the bitterness of beverages.

The above-mentioned correlation patterns can be confirmed by the correlation heatmap (Figure 4F), which provides the two major classes and three subclasses. In this correlation matrix, the highly correlated features were grouped in deep red and the components in light green revealed negative or non-significant correlations. The E. tongue taste traits, including the SWS-sweetness, the STS-saltiness, the BRS_bitterness, and the volatiles, including methyl formate, butane, butylate, and 2-methylpentane, were included in major class I, while displaying other components and characteristics in class II. The umami taste and the sourness fall into the class that highly correlated to the presence of flavor metabolites, phenolic components, and flavonoid components. According to these findings, the RSGW beverages apparently suppressed the undesirable bitter and salty tastes due to the presence of aroma metabolites.

### 3.6. Clustering Analysis and Principal Component Analysis of Metabolites in Hot and Cold Beverages

In order to comprehensively understand the flavor metabolites in different infusion loads and types of beverages, multivariate and chemometric analyses were applied to illustrate the clustering heatmap (Figure 5A) and the principal component analysis (PCA) (Figure 5B). The clustering heatmap provided the component of each feature in different samples by arranging stronger color intensities (from light green to red) in order to increase the level of components. As shown in Figure 5B, the PCA biplot differentiated the hot and cold beverage samples with respect to their bioactive components, their antioxidant capacities, their amino acids, their volatiles, and their taste traits by dividing them into two PC (F1: 62.96% and F2: 21.50%). In particular, the cold beverages were allocated on the positive side of PC1, and the hot beverages were allocated on the negative side. All of the amino acids and bioactive components, as well as most of the volatiles, were established in PC1, with the umami and sourness taste traits, while negatively positioning the other taste traits (saltiness, bitterness, and sweetness) in the opposite PC2. The considerably higher contribution of hydrocarbons, such as 8-methylpentadecane, 2-methylpentane, and butane, and esters, such as butylate and methyl formate, in the hot beverages were clearly grouped as HB_2, HB_1, and HB_3 in PC2. CB_3 pinpointed the sourness taste, due to the presence of the most volatile components, including the sour amino acid (Ile). The umami taste was clearly established in the CB_2 sample, with the existence of most of the amino acids, as well as TPC, TFC, DPPH, and TEAC. This demonstrates a distinct classification in the composition of flavor metabolites and nutraceutical characteristics with respect to the possible taste that is sensed by electronic sensors between hot and cold beverages.

## 4. Conclusions

These findings have clearly demonstrated the nutraceutical characteristics, including the total flavonoid and phenolic contents, as well as the antioxidant capacities (DPPH and TEAC), 16 amino acids, 24 volatiles, and 5 taste attributes that can be detected by sensors in hot beverages (hot infusion) and cold beverages (boiled and cold serving) with different loadings. The correlation pattern and heatmap showed a remarkable relationship between the taste and the flavor compounds. The multivariate statistical assessments, such as clustering analysis and PCA, clearly divided the key metabolites that were attributed to the different tastes in the hot and cold beverages. In summary, the cold beverages exhibited a major abundance of volatile and amino acid components with higher nutraceutical properties than the hot beverages and the associated sourness and umami taste traits. Based on these observations, the serving type of the wheat beverage exerted significant differences in the quality characteristics, the taste, and the flavor profiles. Further studies are needed in order to explore sensory characterization using consumer-based methodologies.

## Figures and Tables

**Figure 1 foods-11-03099-f001:**
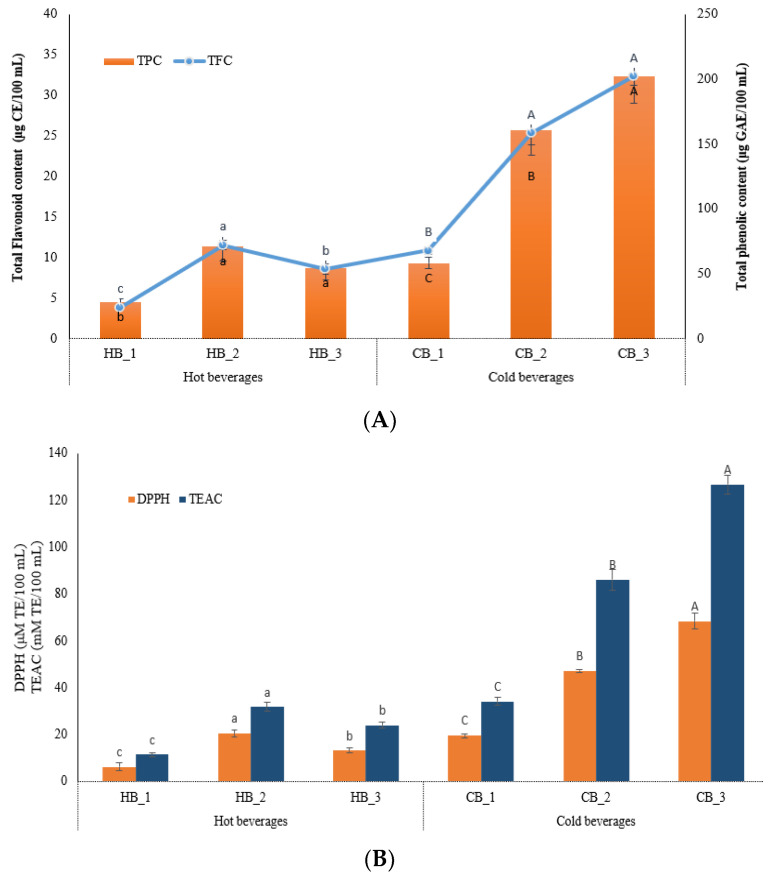
Nutraceutical characteristics of hot- and cold-brewed beverages. (**A**) Bioactive components, including total flavonoid content (TFC) (μg CE/100 mL) and total phenolic content (TPC) (μg GAE/100 mL), (**B**) antioxidant activities, including DPPH radical scavenging activity (μM TE/100 mL) and TEAC (mM TE/100 mL). HB_1, hot beverage 0.8 g/100 mL; HB_2, hot beverage 2 g/100 mL; HB_3, hot beverage 4 g/100 mL; CB_1, cold beverage 25 g/L; CB_2, cold beverage 50 g/L; CB_3, cold beverage 75 g/L; CE, catechin equivalent; GAE, gallic acid equivalent; TE, Trolox equivalent. The different small letters indicate significant difference between the hot-brewed beverages and the different capital letters indicate significant difference between the cold-brewed beverages (*p* < 0.001).

**Figure 2 foods-11-03099-f002:**
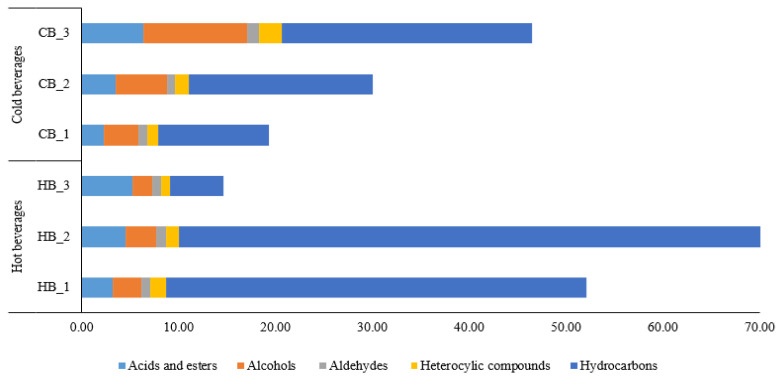
Total volatile components of hot- and cold-brewed beverages. HB_1, hot beverage 0.8 g/100 mL; HB_2, hot beverage 2 g/100 mL; HB_3, hot beverage 4 g/100 mL; CB_1, cold beverage 25 g/L; CB_2, cold beverage 50 g/L; CB_3, cold beverage 75 g/L.

**Figure 3 foods-11-03099-f003:**
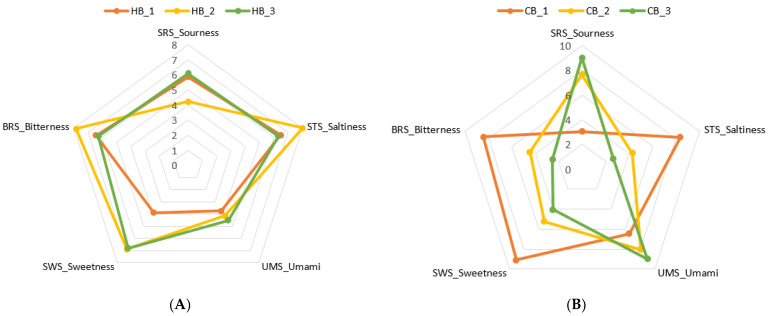
Spider chart for the taste profile of hot beverages (**A**) and a spider chart for the taste profile of cold beverages (**B**). HB_1, hot beverage 0.8 g/100 mL; HB_2, hot beverage 2 g/100 mL; HB_3, hot beverage 4 g/100 mL; CB_1, cold beverage 25 g/L; CB_2, cold beverage 50 g/L; CB_3, cold beverage 75 g/L.

**Figure 4 foods-11-03099-f004:**
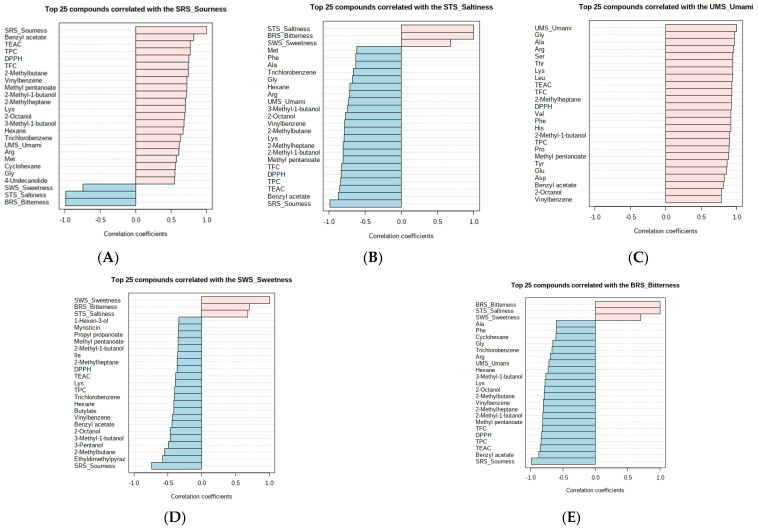
Correlation analysis between aroma metabolites and characteristics of hot and cold beverages. The pattern of the top 25 compounds correlated with the sourness (**A**), saltiness (**B**), umami (**C**), sweetness (**D**), and bitterness (**E**), and the Pearson correlation heatmap (**F**). Ala, alanine; Arg, arginine; Asp, aspartic acid; Glu, glutamic acid; Gly, glycine; His, histidine; Ile, isoleucine; Leu, leucine; Lys, lysine; Met, methionine; Phe, phenylalanine; Pro, proline; Ser, serine; Thr, threonine; Tys, tyrosine; Val, valine.

**Figure 5 foods-11-03099-f005:**
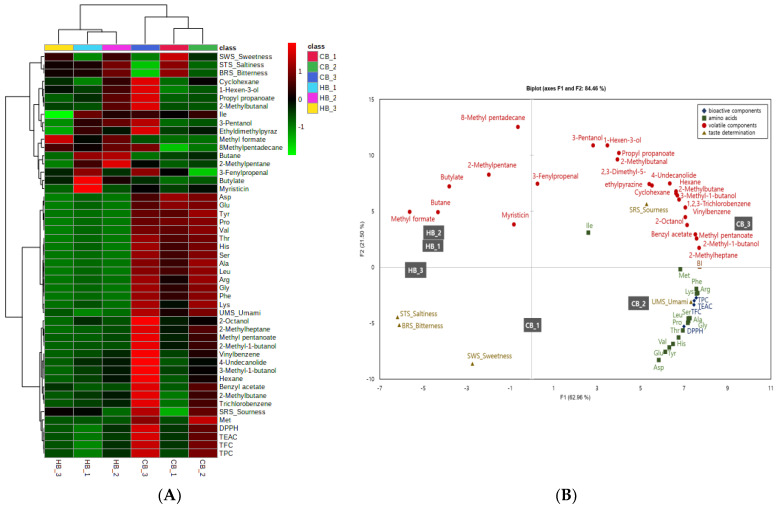
Clustering heatmap (**A**) and principal component analysis (PCA) (**B**) of nutraceutical properties, amino acid compositions, volatile compounds, and the taste profile in hot and cold beverages. HB_1, hot beverage 0.8 g/100 mL; HB_2, hot beverage 2 g/100 mL; HB_3, hot beverage 4 g/100 mL; CB_1, cold beverage 25 g/L; CB_2, cold beverage 50 g/L; CB_3, cold beverage 75 g/L; Ala, alanine; Arg, arginine; Asp, aspartic acid; Glu, glutamic acid; Gly, glycine; His, histidine; Ile, isoleucine; Leu, leucine; Lys, lysine; Met, methionine; Phe, phenylalanine; Pro, proline; Ser, serine; Thr, threonine; Tys, tyrosine; Val, valine.

**Table 1 foods-11-03099-t001:** Amino acid profile of hot- and cold-brewed beverages.

Amino Acids (mg/100 g) ^(1)^	Hot Beverages			Cold Beverages		
	HB_1	HB_2	HB_3	CB_1	CB_2	CB_3
Ala	3.29 ± 0.39 ^a^	2.75 ± 0.06 ^ab^	2.61 ± 0.17 ^c^	10.21 ± 0.95 ^B^	14.32 ± 0.34 ^A^	13.50 ± 0.71 ^A^
Arg	2.22 ± 0.46 ^a^	1.71 ± 0.08 ^a^	1.62 ± 0.14 ^a^	6.10 ± 0.96 ^B^	10.33 ± 1.42 ^A^	10.53 ± 0.90 ^A^
Asp	2.78 ± 0.42 ^a^	2.36 ± 0.23 ^a^	2.18 ± 0.21 ^a^	8.09 ± 0.29 ^A^	7.46 ± 0.52 ^A^	6.42 ± 0.17 ^B^
Glu	1.38 ± 0.28 ^a^	1.05 ± 0.06 ^a^	1.00 ± 0.08 ^a^	4.96 ± 0.30 ^A^	4.77 ± 0.13 ^A^	4.21 ± 0.18 ^B^
Gly	1.29 ± 0.25 ^a^	1.01 ± 0.06 ^a^	1.02 ± 0.08 ^a^	3.92 ± 0.67 ^A^	5.54 ± 0.70 ^A^	5.98 ± 0.55 ^A^
His	1.02 ± 0.24 ^a^	0.69 ± 0.03 ^a^	0.72 ± 0.03 ^a^	2.96 ± 0.54 ^A^	3.57 ± 0.84 ^A^	4.03 ± 0.72 ^A^
Ile	1.90 ± 0.21 ^a^	1.57 ± 0.21 ^ab^	0.24 ± 0.02 ^c^	1.44 ± 0.13 ^A^	1.66 ± 0.11 ^A^	1.67 ± 0.07 ^A^
Leu	0.65 ± 0.00 ^a^	0.45 ± 0.01 ^a^	0.43 ± 0.01 ^b^	2.78 ± 0.48 ^A^	3.25 ± 0.28 ^A^	3.18 ± 0.34 ^A^
Lys	0.95 ± 0.01 ^a^	0.61 ± 0.05 ^a^	0.61 ± 0.05 ^b^	1.65 ± 0.39 ^B^	3.36 ± 0.45 ^A^	3.52 ± 0.29 ^A^
Met	ND	ND	ND	ND	0.37 ± 0.28 ^A^	0.30 ± 0.07 ^A^
Phe	0.23 ± 0.04 ^a^	0.15 ± 0.01 ^a^	0.15 ± 0.01 ^b^	1.08 ± 0.41 ^A^	1.55 ± 0.32 ^A^	1.64 ± 0.25 ^A^
Pro	4.05 ± 0.58 ^a^	3.26 ± 0.43 ^a^	2.98 ± 0.54 ^a^	14.8 ± 1.80 ^A^	15.59 ± 1.23 ^A^	14.23 ± 1.47 ^A^
Ser	2.21 ± 0.38 ^a^	1.77 ± 0.06 ^a^	1.81 ± 0.11 ^a^	8.36 ± 1.84 ^A^	10.54 ± 1.42 ^A^	11.31 ± 1.00 ^A^
Thr	0.69 ± 0.12 ^a^	0.54 ± 0.03 ^ab^	0.50 ± 0.01 ^b^	2.24 ± 0.31 ^B^	2.52 ± 0.29 ^AB^	2.82 ± 0.28 ^A^
Tyr	0.32 ± 0.04 ^a^	0.24 ± 0.00 ^b^	0.24 ± 0.02 ^b^	1.56 ± 0.29 ^A^	1.97 ± 0.61 ^A^	1.64 ± 0.26 ^A^
Val	1.01 ± 0.21 ^b^	0.73 ± 0.05 ^a^	0.69 ± 0.07 ^a^	3.67 ± 0.48 ^A^	4.18 ± 0.12 ^A^	3.94 ± 0.34 ^A^

HB_1, hot beverage 0.8 g/100 mL; HB_2, hot beverage 2 g/100 mL; HB_3, hot beverage 4 g/100 mL; CB_1, cold beverage 25 g/L; CB_2, cold beverage 50 g/L; CB_3, cold beverage 75 g/L. ^(1)^ Ala, alanine; Arg, arginine; Asp, aspartic acid; Glu, glutamic acid; Gly, glycine; His, histidine; Ile, isoleucine; Leu, leucine; Lys, lysine; Met, methionine; Phe, phenylalanine; Pro, proline; Ser, serine; Thr, threonine; Tys, tyrosine; Val, valine. ND, Not detected. The values are expressed as the mean of three replicates ± SD. The different small letters indicate significant differences between the hot beverages, and the different capital letters indicate significant differences between the cold beverages (*p* < 0.05).

**Table 2 foods-11-03099-t002:** Volatile profile of roasted germinated wheat beverages by E-nose (Peak area × 10^3^).

Compounds	Odor Description	Hot Beverages			Cold Beverages		
		HB_1	HB_2	HB_3	CB_1	CB_2	CB_3
**Acids and esters (5)**							
Methyl formate	Agreeable, Fruity	1.32 ± 0.12 ^c^	2.64 ± 0.45 ^b^	3.77 ± 0.20 ^a^	0.17 ± 0.00 ^A^	0.18 ± 0.02 ^A^	0.50 ± 0.70 ^A^
Propyl propanoate	Sweet, Fruity	0.22 ± 0.03 ^a^	0.30 ± 0.11 ^a^	0.18 ± 0.02 ^a^	0.18 ± 0.00 ^B^	0.19 ± 0.02 ^B^	0.37 ± 0.07 ^A^
Methyl pentanoate	Nutty, Sweet	0.46 ± 0.06 ^a^	0.71 ± 0.06 ^a^	0.46 ± 0.01 ^a^	1.20 ± 0.09 ^B^	2.35 ± 0.09 ^B^	4.56 ± 0.95 ^A^
Benzyl acetate	Burnt, Sweet	0.13 ± 0.01 ^a^	0.13 ± 0.01 ^a^	0.15 ± 0.02 ^a^	0.13 ± 0.00 ^B^	0.22 ± 0.01 ^A^	0.29 ± 0.05 ^A^
Butylate	Aromatic	1.04 ± 0.12 ^a^	0.74 ± 0.12 ^b^	0.65 ± 0.07 ^b^	0.54 ± 0.06 ^A^	0.57 ± 0.09 ^A^	0.64 ± 0.19 ^A^
**Alcohols (5)**							
3-Pentanol	Nutty, Sweet	0.52 ± 0.06 ^ab^	0.62 ± 0.23 ^a^	0.18 ± 0.01 ^b^	0.25 ± 0.01 ^B^	0.28 ± 0.02 ^B^	0.77 ± 0.12 ^A^
3-Methyl-1-butanol	Burnt, Bitter	0.19 ± 0.04 ^a^	0.26 ± 0.12 ^a^	0.12 ± 0.02 ^a^	0.18 ± 0.01 ^B^	0.30 ± 0.01 ^B^	0.73 ± 0.11 ^A^
2-Methyl-1-butanol	Alcoholic, Sweet	0.55 ± 0.11 ^a^	0.62 ± 0.11 ^a^	0.36 ± 0.04 ^a^	1.47 ± 0.13 ^B^	2.67 ± 0.02 ^B^	5.27 ± 0.93 ^A^
1-Hexen-3-ol	Green	0.32 ± 0.04 ^a^	0.50 ± 0.04 ^a^	0.38 ± 0.03 ^a^	0.22 ± 0.01 ^B^	0.28 ± 0.01 ^B^	0.72 ± 0.20 ^A^
2-Octanol	Aromatic, Walnut	1.39 ± 0.09 ^a^	1.14 ± 0.09 ^a^	0.99 ± 0.05 ^a^	1.47 ± 0.10 ^B^	1.73 ± 0.20 ^B^	3.28 ± 0.28 ^A^
**Aldehydes (2)**							
2-Methylbutanal	Almond, Burnt	0.20 ± 0.03 ^a^	0.33 ± 0.13 ^a^	0.21 ± 0.01 ^a^	0.20 ± 0.01 ^B^	0.20 ± 0.01 ^B^	0.43 ± 0.05 ^A^
3-Fenylpropenal	Sweet	0.77 ± 0.09 ^a^	0.71 ± 0.03 ^a^	0.72 ± 0.04 ^a^	0.73 ± 0.05 ^A^	0.67 ± 0.02 ^A^	0.77 ± 0.11 ^A^
**Heterocyclic compounds (3)**							
2,3-Dimethyl-5-ethylpyrazine	Burnt	0.61 ± 0.19 ^a^	0.44 ± 0.28 ^ab^	0.09 ± 0.01 ^b^	0.34 ± 0.02 ^B^	0.43 ± 0.03 ^B^	1.00 ± 0.23 ^A^
Myristicin	Balsamic, Mild	0.11 ± 0.10 ^a^	0.04 ± 0.03 ^a^	0.04 ± 0.03 ^a^	0.05 ± 0.03 ^A^	0.06 ± 0.04 ^A^	0.06 ± 0.05 ^A^
4-Undecanolide	Fruity	0.85 ± 0.07 ^a^	0.88 ± 0.17 ^a^	0.79 ± 0.13 ^a^	0.79 ± 0.08 ^B^	0.91 ± 0.17 ^AB^	1.27 ± 0.23 ^A^
**Hydrocarbons (9)**							
Butane	Pungent, Strong	16.99 ± 1.97 ^a^	18.66 ± 12.72 ^a^	0.77 ± 0.07 ^a^	3.02 ± 0.39 ^B^	4.82 ± 0.32 ^A^	0.16 ± 0.23 ^C^
2-Methylbutane	Pleasant	2.25 ± 0.11 ^a^	1.38 ± 0.47 ^a^	1.12 ± 0.16 ^a^	0.99 ± 0.06 ^C^	1.74 ± 0.08 ^B^	2.43 ± 0.44 ^A^
2-Methylpentane	-	22.36 ± 2.49 ^b^	40.38 ± 3.98 ^a^	0.80 ± 0.05 ^c^	4.55 ± 0.51 ^C^	7.59 ± 0.31 ^B^	14.29 ± 1.10 ^A^
Hexane	Alkane, Etheral	0.31 ± 0.02 ^a^	0.46 ± 0.25 ^a^	0.21 ± 0.05 ^a^	0.22 ± 0.03 ^B^	0.59 ± 0.05 ^B^	1.37 ± 0.50 ^A^
Cyclohexane	Chloroform	0.14 ± 0.03 ^a^	0.52 ± 0.33 ^a^	0.32 ± 0.01 ^a^	0.19 ± 0.06 ^B^	0.43 ± 0.01 ^B^	0.89 ± 0.20 ^A^
2-Methylheptane	-	0.05 ± 0.04 ^a^	0.08 ± 0.03 ^a^	0.03 ± 0.04 ^a^	0.17 ± 0.01 ^C^	0.34 ± 0.01 ^B^	0.56 ± 0.09 ^A^
Vinylbenzene	Roast, Aromatic	1.62 ± 0.14 ^a^	2.07 ± 0.59 ^a^	1.44 ± 0.03 ^a^	1.77 ± 0.16 ^B^	2.82 ± 0.13 ^B^	5.13 ± 0.83 ^A^
1,2,3-Trichlorobenzene	-	0.39 ± 0.04 ^a^	0.41 ± 0.07 ^a^	0.38 ± 0.02 ^a^	0.36 ± 0.05 ^B^	0.47 ± 0.03 ^AB^	0.56 ± 0.12 ^A^
8-Methyl pentadecane	-	0.35 ± 0.11 ^a^	0.45 ± 0.19 ^a^	0.41 ± 0.02 ^a^	0.10 ± 0.04 ^A^	0.19 ± 0.11 ^A^	0.47 ± 0.33 ^A^

HB_1, hot beverage 0.8 g/100 mL; HB_2, hot beverage 2 g/100 mL; HB_3, hot beverage 4 g/100 mL; CB_1, cold beverage 25 g/L; CB_2, cold beverage 50 g/L; CB_3, cold beverage 75 g/L. The values are expressed as the mean of three replicates ± SD. The different small letters indicate significant differences between the hot beverages, and the different capital letters indicate significant differences between the cold beverages (*p* < 0.05).

## Data Availability

Not applicable.

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
