# Peer review of "Comparative Flavor Profile of Roasted Germinated Wheat (Triticum aestivum L.) Beverages Served Hot and Cold Using Electronic Sensors Combined with Chemometric Statistical Analysis"

_foods, 2022, doi:10.3390/foods11193099_

Round 1

Reviewer 1 Report

This article investigates the comparative taste characteristics of hot and cold roasted sprouts with the help of electronic nose and electronic tongue. The results obtained are very interesting but the manuscript needs revision.

The comments were listed as following:

Line 82: How many hours does overnight mean?

Line 134-139: The working steps of the electronic nose are further explained. Specify the time of cleaning and injection of samples.

How many sensors did the electronic nose have? The names and specifications of the sensors should be noted.

It is better to add the Discussion section and expand that section with previous research.

It is better for authors to reduce the number of their references. There are many references of authors in the references section.

Reviewer 2 Report

1-Moved from 3.5 to 3.7. Line 326-410 (3.6 forgotten)

2-Could this title get any better?

‘Comparative flavor profile of roasted germinated wheat beverages served hot and cold using electronic sensors (Electronic Nose &Tongue) combined with chemometric statistical analysis’

3-The definition of hot and cold roasted germinated wheat (Triticum aestivum L.) beverages in the title can be given in detail in the summary and content.

4-Line 133-missing word-Volatile Compounds

5-Line 169 to 172

‘Generally, hot beverages showed lower TFC and TPC contents as well as antioxidant activities compared to cold samples  due to the amount of solvent used for brewing. In cold beverages, it can be clearly seen that the greater the concentration of RSGW in the preparation, the higher the level of bioactive components and antioxidant capacities.’

According to the statement given here, shouldn't boiling be done in the production of cold drinks?

‘For the preparation of  cold beverages (CB), 20, 50, and 75 g of RSGW was mixed with 1 L water, boiled for 30  min, and cooled at room temperature for 10 min, then labelled as CB_1, CB_2, and CB_3,  respectively.’

6-In the introduction, the difference between wheat and germinated wheat could be expressed more scientifically.

7- A nutraceutical is defined as any substance that is a food or part of a food and provides medical or health benefits, including the prevention and treatment of disease.

https://www.sciencedirect.com/topics/agricultural-and-biological-sciences/nutraceutical

Line 165- What do you call bioactive components? It is not specified which substances.

8- The subject of electronic sensors could have been given in more detail in the introduction. Studies could be added. (For those who don't know the subject or read it for the first time.)

9-The product production flow chart could have been given with a photograph. (It would be nice if it was made to be more striking and memorable.)

10-More literature can be added.

-What are the substances formed by germination in wheat?

-Why did you prepare hot and cold drinks that way?

-What is checked in food with the electronic nose and tongue?

-If there are similar studies, what has been done? Otherwise, you should emphasize this better if you did it first.

-Functionality is associated with health. So what kind of health benefits will there be when this hot and cold beverage is consumed?

-Also, is the perception of taste perceived in the same way by people?

11-Nice work.

Round 2

Reviewer 1 Report

The corrections are done well and I have no other comments.